# Development of Acid Hydrolysis-Based UPLC–MS/MS Method for Determination of *Alternaria* Toxins and Its Application in the Occurrence Assessment in Solanaceous Vegetables and Their Products

**DOI:** 10.3390/toxins15030201

**Published:** 2023-03-06

**Authors:** Hongxia Tang, Wei Han, Shaoxiang Fei, Yubo Li, Jiaqing Huang, Maofeng Dong, Lei Wang, Weimin Wang, Ying Zhang

**Affiliations:** 1Pesticide Safety Evaluation Research Center, Shanghai Academy of Agricultural Sciences, Shanghai 201403, China; 2Key Laboratory for Safety Assessment (Environment) of Agricultural Genetically Modified Organisms, Ministry of Agriculture and Rural Affairs, Institute for Agro-Food Standards and Testing Technology, Shanghai Academy of Agricultural Sciences, Shanghai 201403, China; 3School of Resources and Environment, Northeast Agricultural University, Harbin 150030, China

**Keywords:** *Alternaria* toxins, modified analytical method, solanaceous vegetables, solanaceous vegetable products, UPLC–MS/MS

## Abstract

In this work, we proposed an acid hydrolysis-based analytical method for the detection of *Alternaria* toxins (ATs) in solanaceous vegetables and their products with solid-phase extraction (SPE) and ultrahigh-performance liquid chromatography–tandem mass spectrometry (UPLC–MS/MS). This study was the first to reveal that some compounds in the eggplant matrix bind to altenusin (ALS). Validation under optimal sample preparation conditions showed that the method met the EU criteria, exhibiting good linearity (R^2^ > 0.99), matrix effects (−66.6–−20.5%), satisfying recovery (72.0–107.4%), acceptable precision (1.5–15.5%), and satisfactory sensitivity (0.05–2 µg/kg for limit of detection, 2–5 µg/kg for limit of quantification). Out of 393 marketed samples, only 47 samples were detected, ranging from 0.54–806 μg/kg. Though the occurrence ratio (2.72%) in solanaceous vegetables could be negligible, the pollution status in solanaceous vegetable products was much more serious, and the incidences were 41.1%. In the 47 contaminated samples, the incidences were 4.26% for alternariol monomethyl ether (AME), 6.38% for alternariol (AOH) and altenuene (ALT), 42.6% for tentoxin (TEN), and 55.3% for tenuazonic acid (TeA).

## 1. Introduction

As secondary metabolites of *Alternaria* strains, *Alternaria* toxins (ATs) are readily produced during unfavorable climatic conditions, insect attacks, and the transport, processing, and storage of agricultural products, and are widely found in various foods, posing great potential risks to human and animal health due to their carcinogenic, teratogenic, and cytotoxic properties [1,2,3,4]. Although more than 70 ATs had been confirmed to be poisonous, contamination and dietary safety assessments had focused on only some ATs, including alternariol (AOH), alternariol monomethyl ether (AME), altenuene (ALT), altenusin (ALS), tentoxin (TEN), and tenuazonic acid (TeA) [5,6,7]. Previous studies had revealed that AT contamination in vegetables [8], fruits [8,9,10], tomato products [11], wine [12,13], cereal grains [14,15], dried fruits [16], and juices [12] was widespread, and a high incidence had been observed in processed products, ranging from 62–100% [11,12,17,18,19]. Among these, vegetables and their products were also considered to be some of the most highly AT-contaminated foods [20]. Therefore, due to the considerable consumption of these products and their high susceptibility to *Alternaria* infection, the occurrence of ATs on vegetables and their products in China had been studied, revealing the contamination status and providing data regarding dietary safety risks, which have rarely been reported in China compared to other countries.

As described in these reports, many analytical methods have been proposed for the detection of ATs or ATs and other mycotoxins in various food matrixes. Compared with gas chromatography‒mass spectrometry (GC‒MS) and liquid chromatography–diode array detection (LC–DAD), liquid chromatography–tandem mass spectrometry (LC–MS/MS) was the best choice for detecting ATs due to its excellent sensitivity and high efficiency [2,6,8,9,21,22,23]. However, for sample pretreatment, acetonitrile or formic acid-acetonitrile were the most common extraction solvents following solid-phase extraction (SPE), the QuEChERS (quick, easy, inexpensive, effective, rugged, and safe) method, or direct injection with the isotope internal standard method [4,19,24,25]. Given the availability of different matrixes, SPE cartridges (HLB) and adsorption materials (GCB and C_18_) had been optimized and utilized to clean up the samples [1,5,10,22]. However, among these studies, only a few had achieved the simultaneous detection of six ATs (AOH, AME, ALT, ALS, TEN, and TeA) in wine [12], herbs [6], and juices [12] via the QuEChERS method or direct analysis. It is well known that the adjustment and optimization of the sample pretreatment are necessary for accuracy when the analytes and matrixes have been changed. As shown in this work, the existing analytical methods would not be applicable because of the binding residue of ALS. Therefore, a new method needed to be established to ensure the accuracy of the monitoring results.

Among the various vegetable species, including cucumber, tomato, eggplant, and pepper, solanaceous vegetables undoubtedly play an important role in vegetable production, household consumption, and processed products [26]. Solanaceous vegetables are well known to be susceptible to *Alternaria* infection, which occurs more widely in processed products [11]. Therefore, tomato and its products are the main vegetables used for AT contamination monitoring and safety evaluation. However, in China, relevant reports have tended to focus more on the development of methods than on the investigation of contamination status, and due to differences in consumption habits, it has been more realistic and in line with the requirements for food safety assessment to carry out pollution monitoring of ATs in the main foods consumed in China, such as chili paste and cherry tomato. In this work, using solanaceous vegetables and their products as the subject, an AT occurrence assessment was first carried out to determine the contamination level in Shanghai, which is often overlooked and has become a monitoring gap in food safety evaluations in China.

Based on validation via the reported analytical methods, a new and reliable analytical method was developed and validated for the detection of six ATs (AOH, AME, ALT, ALS, TEN, and TeA) in solanaceous vegetables and their products using acid hydrolysis, SPE, and ultrahigh-performance liquid chromatography–tandem mass spectrometry (UPLC–MS/MS). To ensure the reliability and credibility of the research results, 939 commercial samples were collected, including chili paste, eggplant, ketchup, pepper, and tomato samples. Finally, a survey of AT contamination in solanaceous vegetables and their production in the Shanghai area was proposed, representing the first time that the regional contamination levels of six ATs in solanaceous vegetables and their products had been studied in China.

## 2. Results and Discussion

### 2.1. Optimization of Extraction

#### 2.1.1. Validation via Reported Methods

By using previously reported methods [10,16], validation was carried out via extraction with acetonitrile, but the matrix was replaced with chili paste, eggplant, and tomato. As shown in Figure 1A, although the results showed that most recoveries were acceptable (74.3–104.7%), the value for ALS in eggplant was only 16.2% when the spiked level was 1000 µg/kg. Then, regardless of whether the volume of the extraction solution was expanded, or whether the extraction solvent (such as acetone and methanol) was adjusted and some substances (such as formic acid and ammonium acetate) were added, there was no obvious improvement in the recovery of ALS in eggplant.

In line with the structural formula of ALS, there were two hydroxyl groups and one carboxyl group, and the bound residues may have formed because of the presence of alkaline compounds in eggplant; in comparison, good results were obtained in tomato due to its acidity. Additionally, this result was confirmed by spiking 2 mL of ALS standard solution in acetonitrile (1000 µg/L) into 2 g of eggplant. After vortexing for 15 min, the concentration of ALS in acetonitrile was only 380 µg/L, showing the existence of a binding residue. Therefore, as the first step, it was necessary to eliminate the binding residue between ALS and the eggplant matrix to improve the recovery of ALS.

#### 2.1.2. Optimization of Extraction Conditions

According to the relevant literature, hydrochloric acid and calefaction were selected to remove the binding residue, which was a conventional and effective approach used in some analytical methods [23,27,28,29,30]. Focusing on the concentration of hydrochloric acid, heating time, and heating temperature, the optimization of extraction conditions was carried out using the recovery test in 2.0 g of eggplant (1000 µg/kg) and direct analysis after extraction in 8 mL of acetonitrile.

After exposure to 80 °C for 15 min, the effect of the different concentrations of hydrochloric acid (2 mL) on AT recovery was first estimated. The volume of hydrochloric acid solvent was 2 mL, and the concentrations of hydrochloric acid selected were 0, 0.01, 0.05, 0.1, 0.5, 1, 2, and 5 mol/L. Figure 1B showed that the recovery of ALS was closely related to the concentration of hydrochloric acid, but for other ATs, the values were not significantly changed, ranging from 81.1–113.4%. When the concentration of hydrochloric acid was 0.1, 0.5, or 1 mol/L, the recoveries (56.1–68.9%) of ALS were distinctly higher than the values (38.9–55.1%) under other conditions reported for the analytical method, which was a significant improvement over the reported analytical methods. Therefore, at three hydrochloric acid concentrations (0.1, 0.5 and 1 mol/L) and 80 °C, the recoveries were determined while varying the heating time. Figure 1C–E showed the results at the three different hydrochloric acid concentrations. At heating times ranging from 0–60 min, the recoveries of ALT, AME, AOH, and TEN were stable and acceptable, with values of 74.1–111.7%. Although the range (77.8–113.3%) for TeA was satisfactory, the values decreased when increasing the hydrochloric acid concentration or heating time, especially when the heating time was 45 min or 60 min. However, for ALS, the heating time had a significant effect on the results, and all the recoveries increased as the heating time was extended under the three hydrochloric acid concentrations. At hydrochloric acid concentrations of 0.5 mol/L and 1 mol/L and heating times of 30–60 min, the recoveries of ALS were fairly good, ranging from 77.4–94.6%. Therefore, 0.5 mol/L and 30 min were selected as the best hydrochloric acid concentration and heating time, respectively.

Finally, based on the above tests, the heating temperature was also investigated, and five heating temperatures (25, 40, 60, 80, and 90 °C) were included in the experiment. The results (Figure 1F) showed that the recoveries (75.8–115.5%) of ALT, AME, AOH, and TEN were all acceptable and were consistent with the results of previous experiments. For TeA, when the heating temperature was 90 °C, the recovery was only 49.0%, but in the lower temperature conditions, the values were much more stable and acceptable (77.0–108.9%). The recoveries of ALS increased when increasing the heating temperature (25–80 °C), ranging from 12.4% to 87.8%. However, when the heating temperature was 90 °C, similar to TeA, the value decreased to 48.4%, revealing that both ALS and TeA were degraded at higher temperatures or longer heating times.

Therefore, the following sample pretreatment conditions were established: 2 mL of 0.5 mol/L hydrochloric acid, 30 min of heating time, and a heating temperature of 80 °C.

### 2.2. Optimization of Clean-Up Conditions

To ensure the accuracy of the method and based on other papers, acetonitrile was used to extract the target compounds from the hydrochloric acid mixture, and the extraction solvent volume was optimized. With the volume of acetonitrile set to 2, 4, 6, 8, 12, 16, or 20 mL, the recoveries of ATs in eggplant were determined. As shown in Figure 2A, when the volume of acetonitrile was in the range of 4–20 mL, the recoveries of ALS, ALT, AME, AOH, and TEN were 72.2–91.0%, but for TeA, the volume of acetonitrile needed to be at least 8 mL, attaining recoveries of 81.7–87.3%. Therefore, 8 mL was selected as the volume of acetonitrile for the extraction solvent.

As shown in the literature and for the standards, SPE is a powerful clean-up tool in analytical methods [9,13,16,22]. Although some papers have described the application of SPE for the detection of ATs, to date, simultaneous analysis of all six major toxins via SPE has not been reported. Therefore, in this work, we attempted to select a suitable SPE column due to the change in the target object. Eleven kinds of commercial SPE columns were tested. As shown in Figure 2B, after clean-up using the SPE columns and elution with 2 mL of acetonitrile, except for the C_8_ columns, none of the 10 columns could achieve the simultaneous purification of six target compounds, attaining recoveries of 0.2–123.1%, which did not meet the requirements of the EU guidelines (70–120%). However, with the C_8_ columns, the recoveries (77.7–112.7%) of the six ATs were much better and met the requirements of analytical methods. Therefore, the C_8_ columns were selected as the clean-up tool.

### 2.3. Validation Method

For validation, five typical solanaceous vegetables and vegetable products, namely chili paste, eggplant, ketchup, pepper, and tomato, were selected as the evaluation matrixes. Moreover, six ATs were isolated through chromatographic separation (Figure 3). As shown in Table 1, in different concentration ranges and matrixes, the target compounds exhibited a good linear relationship, with correlation coefficients exceeding 0.998. Then, for the MEs, as shown in Figure 4, the values in these matrixes indicated obvious matrix inhibition effects, and the recoveries in unpurified samples and purified samples were −94.7–−47.1% and −66.6–−20.5%, respectively. All the absolute values for the ME were significantly lower than those of the unpurified samples, proving that the proposed method had an obvious impurity removal effect. However, because the values exceeded ±20%, the matrix standard solution was indispensable for accurate quantitative analysis. The accuracy and precision were determined via the spiked recovery test. Table 1 showed that in the five matrixes, all the recoveries and RSDs were acceptable, reaching 72.0–107.4% and below 15.5%, respectively. Calculated according to three signal/noise ratios, the LODs of the six ATs ranged from 0.05–2 μg/kg in the five matrixes. The LOQ of AME was 5 μg/kg, and for other ATs, the LOQ was 2 μg/kg. Therefore, the validation results demonstrated that the proposed analytical method was acceptable and satisfactory, complying with EU criteria [31].

### 2.4. Method Application

In 2019, a total of 939 commercialized samples were randomly collected from supermarkets, online vendors and farmer’s markets in Shanghai, China. The collected samples comprised 33 chili paste, 244 eggplant, 23 ketchup, 450 pepper, and 189 tomato (including cherry tomato) samples. As shown in Table 2, overall, the results showed that only 5.00% were (47/939) positive samples among the collected samples, and except for ALS, the other five target compounds were detected, with levels ranging from 0.54–806 μg/kg. Although the AT contamination range in the solanaceous vegetables was low (only 2.72%; 24/883), the rate in the solanaceous vegetable products was much higher, reaching 41.1% (23/56). Among the samples, the rates of positive samples, from high to low, were as follows: 65.2% (15/23) for ketchup, 24.2% (8/33) for chili paste, 3.70% (7/189) for tomato, 2.46% (6/244) for eggplant, and 2.44% (11/450) for pepper. In general, the contamination rates (2.44–3.70%) in solanaceous vegetables were not significantly different among eggplant, pepper, and tomato, but the values in ketchup and chili paste reached up to 65.2% and 24.2%, respectively, proving that AT contamination in solanaceous vegetable products was greater than that in solanaceous vegetables. Although there had been reports on AT contamination in vegetables, some studies had focused on other foods [5,10,12,13,14]. Among them, some studies in several countries revealed the AT contamination in ketchup, and the positive rates of the ketchup samples were much higher (40–92%), and the pollution level of ATs ranged from 42–814 μg/kg, which was similar to the results of our study [11,18,19,32,33].

In contrast, AOH, AME, ALT, TEN, and TeA were detected in solanaceous vegetables and solanaceous vegetable products, but the degree of contamination varied with the matrix and AT. Among them, TeA and TEN were the two toxins with the highest contamination frequencies, accounting for 55.3% (26/47) and 42.6% (20/47) of the positive samples, respectively. TeA and TEN were recognized as the most widespread of these toxins, present in materials such as ketchup [11,32], wolfberry [5], grain [17], food commodities [34], and grain-based products [14]. As the most frequently detected toxin, the concentrations of TeA in the five matrixes were in the range not detected (ND) to 19.3 μg/kg in chili paste, ND–214 μg/kg in eggplant, ND–337 μg/kg in ketchup, ND-806 μg/kg in pepper, and ND in tomato. The concentrations of TEN in the five matrixes were in the range of ND–0.56 μg/kg in chili paste, ND–6.44 μg/kg in eggplant, ND–2.81 μg/kg in ketchup, ND–13.1 μg/kg in pepper, and ND–6.43 μg/kg in tomato. The TeA-positive samples were solanaceous vegetable products (four chili paste and fourteen ketchup) and related solanaceous vegetables (two eggplant and six pepper samples). This was very different from the result for TEN, as the numbers of TEN-positive samples of solanaceous vegetables and solanaceous vegetable products were fifteen (four eggplant, five pepper, and six tomato samples) and five (two chili paste and three ketchup samples), respectively.

However, for AOH, AME, and ALT, there were very few positive samples, with proportions of 6.38% (3/47), 4.26% (2/47), and 6.38% (3/47), respectively. Additionally, the concentrations were also much lower than those of TeA or TEN, with values of 5.29–20 μg/kg for AOH, 11.9–12.6 μg/kg for AME, and 5.75–10.3 μg/kg for ALT. All the samples positive for AOH and AME were ketchup samples; in contrast, ALT was detected only in solanaceous vegetables (one tomato and one pepper).

Therefore, from the above survey results, AT contamination in solanaceous vegetable products is higher and more widespread than that in solanaceous vegetables. Compared with the reported papers, the rates of positive samples or the pollution level in ketchup were prominent, which was similar to the results of our study [32,33]. The low occurrence rate of *Alternaria* infection may be due to the shorter preservation and transportation time of the commercialized solanaceous vegetables. The other reason may be the concentration factors in the processing of the vegetable products. For example, the dry matter of tomatoes is 5–6%, while in ketchup it is 28–30%; this difference may cause the higher detection rates of ATs in the vegetable products. Anyways, the current situation of severe contamination in solanaceous vegetable products led to concerns regarding food safety, and it prompted us to study the contamination in processed foods.

## 3. Conclusions

In light of the ALS binding residue present in the alkaline matrix, the strategy involving the addition of acid and heating at an optimal temperature and time was proposed. The development method was first established for detecting six ATs in eggplant and was also validated in other solanaceous vegetables and their products for monitoring the six ATs in accordance with the EU criteria [31].

Furthermore, a survey of AT contamination was carried out in Shanghai, China. The experimental results for a total of 939 commercialized samples indicated that the AT levels in solanaceous vegetables were negligible with (2.72%, 24/883), but the incidence in solanaceous vegetable products was much higher (41.1%, 23/56). Among the six toxins detected (AOH, AME, ALT, TEN, and TeA), as the most frequently detected ATs, TEN and TeA were present at similar levels, with a detection rate of 42.6–55.3% in the positive samples, which was much higher than the values for AOH, AME, and ALT (4.26–6.38%). This study was the first to describe the issue of mycotoxin-bound residues in a plant origin matrix, and the proposed analytical method also led to the creation of a new preprocessing approach. Additionally, this study revealed AT contamination in solanaceous vegetables and their products in China, which revealed the high detection rate in the solanaceous vegetable products.

## 4. Materials and Methods

### 4.1. Chemicals, Reagents, and Materials

The AOH standard (95% purity) was purchased from Cayman Chemical (Ann Arbor, MI, USA) and the ALT standard (98% purity) was purchased from Toronto Research Chemicals (North York, ON, Canada). The AME (99.5% purity), TeA (99.5% purity), and ALS (99.4% purity) standards were purchased from AdipoGen Life Sciences (San Diego, CA, USA), and the TEN standard was purchased from LKT Laboratories (Sao Paulo, Brazil). HPLC-grade solvents (acetonitrile and methanol) were purchased from Merck (Darmstadt, Germany). Octylsilane (C_8_) (100 mg, 1 mL), ethylsilane (C_2_) (100 mg, 1 mL), phenyl (PH) (100 mg, 1 mL), primary secondary amine (PSA) (100 mg, 1 mL), graphitized carbon black (GCB) (100 mg, 1 mL), and alumina (Al_2_O_3_) (100 mg, 1 mL) columns were obtained from Shimadzu Corp. (Kyoto, Japan). Aminopropyl (HN_2_) (500 mg, 6 mL) and florisil (500 mg, 3 mL) columns were obtained from Agilent Technologies Inc. (Wilmington, DE, USA). Octadecylsilane (C_18_) (200 mg, 3 mL), hydrophilic–lipophilic balance (HLB) (60 mg, 3 mL), and silica (Si) (1 g, 6 mL) columns were obtained from ANPEL Laboratory Technologies Inc. (Shanghai, China).

### 4.2. Sample Preparation

Approximately 2.0 g of homogenized sample were weighed into 50 mL polytetrafluoroethylene (PTFE) centrifuge tubes. After adding 2 mL of 0.5 mol/L hydrochloric acid solvent, the tube was heated in an 80 °C water bath for 30 min. Then 8 mL of acetonitrile were added to the tube at room temperature. After shaking vigorously for 3 min, 2 g of NaCl were added to the tube, and the tube was shaken for 30 s and centrifuged at 4500 rpm for 5 min in preparation for cleanup.

The supernatant was transferred into the SPE cartridge for the cleanup. In the SPE cleanup procedure, the C8 cartridge was pre-eluted with 1 mL of acetonitrile, then 4 mL of the extraction solution were loaded into the C_8_ cartridges, and 2 mL of acetonitrile were loaded onto the cartridge for elution. The collected eluate was dried in a 60 °C water bath under nitrogen. The sample was redissolved in 1 mL of acetonitrile and passed through a 0.22 µm PTFE filter for analysis.

### 4.3. Apparatus and UPLC–MS/MS Conditions

Detection was performed with a UPLC‒MS/MS instrument (triple–quadrupole mass spectrometer) from Shimadzu (Kyoto, Japan) equipped with an electrospray ionization source. Separation of the ATs was achieved using a Waters ACQUITY UPLC BEH C18 (2.1 mm × 50 mm, 1.7 µm) column (Waters, Milford, MA, USA). The injection volume, column temperature, and flow rate were set at 5 μL, 40 °C, and 0.4 mL/min, respectively. The mobile phase included acetonitrile (A) and 0.1% formic acid water, with the following elution gradient: 0–0.5 min (10% A), 0.5–1.5 min (10–90% A), 1.5–3.0 min (90% A), 3.0–3.5 min (90–10% A), and 3.5–5.0 min (10% A).

The low-energy collision dissociation tandem–mass spectrometric analysis (CID–MS/MS) conditions were set using the following parameters: 1.5 mL/min nebulizing gas flow (99.9%, N_2_), 400 °C heat block temperature, 15 mL/min drying gas flow (99.9%, N_2_), 3.5 kV interface voltage, and 230 kPa collision-induced dissociation gas pressure (99.999%, Ar). After automatic optimization, the MS parameters of the ATs were determined and are shown in Table 3. Quantification was performed using multiple reaction monitoring (MRM) of selected precursor ion → product ions transitions.

### 4.4. Method Validation

As an essential step, validation of the analytical method was carried out by determining a series of parameters, including the linear curve, matrix effect (ME), accuracy, precision (relative standard deviations, RSDs), limit of detection (LOD), and limit of quantitation (LOQ). A series of standard solutions with five concentrations were prepared, and the linear curve was described using the concentrations and the instrument response. The ME was obtained via the following formula: ME = (the slope of calibration curves in the matrix—the slope of calibration curves in solvent)/the slope of calibration curves in solvent × 100%. If the ME was within ±20%, it could be ignored; otherwise, there was an obvious ME, and a matrix standard solution was necessary to ensure the accuracy of the analytical methods [35]. After spiking with three different levels and five repetitions in five matrixes (chili paste, eggplant, ketchup, pepper, and tomato), the accuracy and precision of the proposed method were determined. The LOD was the concentration corresponding to three times the signal/noise ratio [7], and the LOQ was considered the lowest spiked level [36].

## Figures and Tables

**Figure 1 toxins-15-00201-f001:**
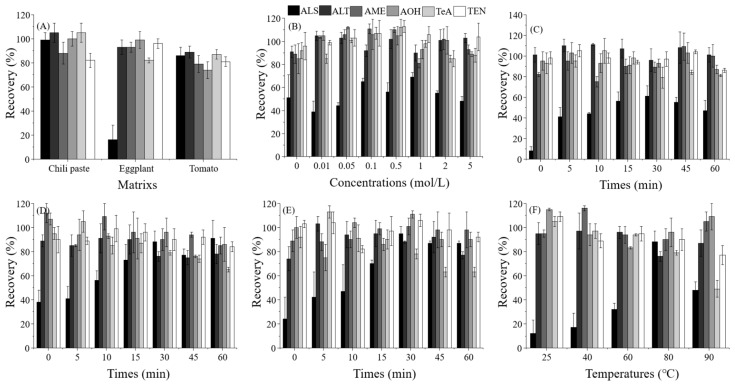
Optimization of the extraction conditions (1000 μg/kg, *n* = 5). (**A**) Recoveries of *Alternaria* toxins from chili paste, eggplant, and tomato by extraction with acetonitrile. (**B**) Optimization of the hydrochloric acid concentration at 80 °C for 15 min in eggplant. (**C**) Optimization of the heating time with 0.1 mol/L hydrochloric acid at 80 °C in eggplant. (**D**) Optimization of the heating time with 0.5 mol/L hydrochloric acid at 80 °C in eggplant. (**E**) Optimization of the heating time with 1 mol/L hydrochloric acid at 80 °C in eggplant. (**F**) Optimization of the heating temperatures with 0.5 mol/L hydrochloric acid for 30 min in eggplant.

**Figure 2 toxins-15-00201-f002:**
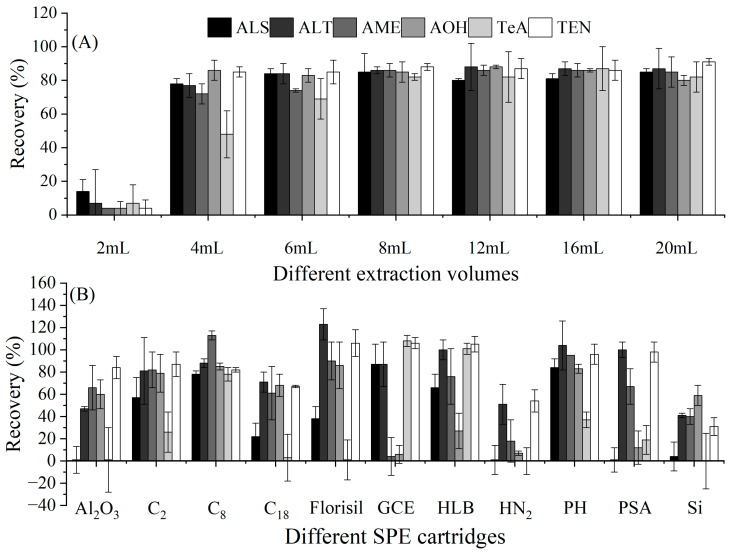
Optimization of the extraction and SPE columns in eggplant (1000 μg/kg, *n* = 3). (**A**) Optimization of the extraction volume. (**B**) Selection of the SPE column.

**Figure 3 toxins-15-00201-f003:**
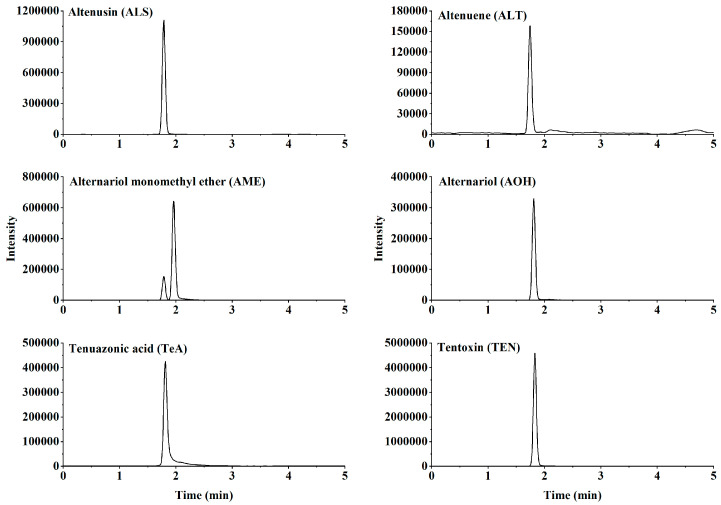
Chromatograms of *Alternaria* toxins in the mixed standard solution. The concentration is 100 μg/L.

**Figure 4 toxins-15-00201-f004:**
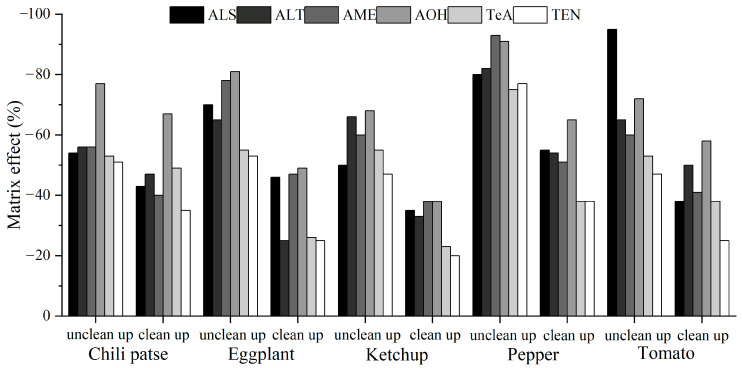
Matrix effects of solanaceous vegetables and their products.

**Table 1 toxins-15-00201-t001:** Method validation for *Alternaria* toxins in solanaceous vegetables and their products.

Compound	Matrix	Linear Range (μg/L)	Calibration Curve	Correlation Coefficient (R^2^)	LOD (μg/kg)	Average Recovery Rate (%) (RSD (%)) (*n* = 6)		
						2 μg/kg	10 μg/kg	1000 μg/kg
ALS	Chili paste	1–1000	y = 4802x + 14064	1.000	0.5	77.2 (5.7)	76.0 (2.6)	82.8 (2.8)
	Eggplant	1–1000	y = 4548x + 39771	0.999	0.5	75.2 (3.4)	81.0 (8.2)	75.6 (3.0)
	Ketchup	1–1000	y = 5456x + 26143	1.000	0.5	75.6 (3.6)	72.0 (2.8)	80.0 (6.4)
	Pepper	1–1000	y = 3766x + 21551	1.000	0.5	72.4 (2.3)	76.0 (4.5)	81.6 (4.6)
	Tomato	1–1000	y = 5304x + 47966	0.999	0.5	76.8 (3.4)	76.6 (3.5)	82.8 (3.4)
ALT	Chili paste	1–1000	y = 1400x + 5590	1.000	0.5	100.4 (15.5)	96.2 (7.5)	91.2 (3.5)
	Eggplant	1–1000	y = 1963x − 1128	0.999	0.5	85.4 (6.7)	85.8 (5.5)	91.2 (4.9)
	Ketchup	1–1000	y = 1780x + 1184	1.000	0.5	97.4 (13.0)	82.6 (5.4)	98.0 (12.5)
	Pepper	1–1000	y = 1219x + 6487	1.000	0.5	96.4 (13.0)	91.0 (4.2)	92.6 (6.6)
	Tomato	1–1000	y = 1329x + 9554	1.000	0.5	103.6 (11.6)	89.0 (4.3)	90.8 (5.9)
AME	Chili paste	5–1000	y = 290x + 2029	0.999	2	93.0 (3.1) ^a^	87.8 (12.8)	92.6 (8.0)
	Eggplant	5–1000	y = 541x + 5632	0.999	2	105.2 (13.7) ^a^	91.0 (1.3)	89.8 (4.1)
	Ketchup	5–1000	y = 627x + 6041	0.999	2	85.4 (10.1) ^a^	86.2 (3.3)	92.8 (10.8)
	Pepper	5–1000	y = 484x + 1091	1.000	2	103.2 (12.2) ^a^	93.2 (6.3)	89.2 (2.9)
	Tomato	5–1000	y = 607x + 13240	0.998	2	87.6 (14.2) ^a^	96.6 (3.2)	96.0 (2.4)
AOH	Chili paste	5–1000	y = 840x + 6452	1.000	2	86.0 (14.1)	93.8 (7.2)	92.2 (3.5)
	Eggplant	1–1000	y = 1291x + 9660	1.000	0.5	86.8 (5.3)	101.4 (2.6)	95.2 (3.6)
	Ketchup	1–1000	y = 1559x + 11605	0.998	0.5	107.4 (4.7)	89.2 (6.6)	95.8 (10.8)
	Pepper	1–1000	y = 875x + 8618	1.000	0.5	85.4 (14.8)	92.6 (5.3)	92.4 (4.2)
	Tomato	1–1000	y = 1056x + 12184	0.999	0.5	84.2 (15.1)	89.8 (7.1)	93.0 (3.8)
TeA	Chili paste	0.5–1000	y = 9007x + 59591	1.000	0.2	73.2 (3.5)	75.4 (4.8)	78.6 (3.6)
	Eggplant	0.1–1000	y = 12988x + 70681	1.000	0.05	74.8 (2.4)	77.2 (2.8)	76.8 (6.4)
	Ketchup	0.1–1000	y = 13465x + 36066	1.000	0.05	75.4 (4.5)	76.2 (3.3)	82.0 (4.6)
	Pepper	0.5–1000	y = 10670x + 22379	1.000	0.2	72.6 (3.3)	75.2 (1.5)	77.8 (3.7)
	Tomato	0.5–1000	y = 10876x + 70959	1.000	0.2	75.8 (2.9)	75.8 (3.4)	81.8 (7.7)
TEN	Chili paste	0.1–1000	y = 12846x + 23279	1.000	0.05	87.6 (8.2)	80.6 (5.5)	96.0 (2.4)
	Eggplant	0.1–1000	y = 14980x + 31438	1.000	0.05	86.8 (1.5)	84.0 (2.4)	93.6 (3.8)
	Ketchup	0.1–1000	y = 15575x + 12590	1.000	0.05	94.0 (11.2)	78.2 (4.2)	96.0 (10.7)
	Pepper	0.1–1000	y = 12366x + 25385	1.000	0.05	90.2 (2.1)	94.6 (4.1)	93.4 (3.5)
	Tomato	0.1–1000	y = 14814x + 26054	1.000	0.05	94.4 (10.6)	92.2 (5.9)	92.4 (3.9)

^a^: The lowest spiked level of AME was 5 μg/kg.

**Table 2 toxins-15-00201-t002:** Contamination by *Alternaria* toxins in solanaceous vegetables and their products in Shanghai, China.

Matrix	N^pos^/N	N^qual^	N^quant^	Avg^quan^ (μg/kg)	Min^quan^ (μg/kg)	Max^quan^ (μg/kg)
Chili paste (N = 33)						
ALS	0/33	0	0	-	-	-
ALT	0/33	0	0	-	-	-
AME	0/33	0	0	-	-	-
AOH	0/33	0	0	-	-	-
TeA	4/33	0	4	12.1	3.44	19.3
TEN	2/33	2	0		-	-
Eggplant (N = 244)						
ALS	0/244	0	0	-	-	-
ALT	0/244	0	0	-	-	-
AME	0/244	0	0	-	-	-
AOH	0/244	0	0	-	-	-
TeA	2/244	0	2	128	41.8	214
TEN	4/244	2	2	4.27	2.10	6.44
Ketchup (N = 23)						
ALS	0/23	0	0	-	-	-
ALT	0/23	0	0	-	-	-
AME	2/23	0	2	12.3	11.9	12.6
AOH	3/23	0	3	8.06	5.75	10.3
TeA	14/23	0	14	85.1	5.61	337
TEN	3/23	2	1	2.81	2.81	2.81
Pepper (N = 450)						
ALS	0/450	0	0	-	-	-
ALT	2/450	0	2	18.2	16.3	20.0
AME	0/450	0	0	-	-	-
AOH	0/450	0	0	-	-	-
TeA	6/450	0	6	157	7.29	806
TEN	5/450	2	3	8.07	2.28	13.1
Tomato (N = 189)						
ALS	0/189	0	0	-	-	-
ALT	1/189	0	1	5.29	5.29	5.29
AME	0/189	0	0	-	-	-
AOH	0/189	0	0	-	-	-
TeA	0/189	0	0	-	-	-
TEN	6/189	4	2	4.58	2.72	6.43

N^pos^/N: Number of positive samples/number of samples analyzed, N^qual^: number of samples above LOD but below LOQ, N^quant^: number of quantified samples, Avg^quan^: mean concentrations of *Alternaria* toxins in quantified samples, Min^quan^: minimum concentrations of *Alternaria* toxins in quantified samples, Max^quan^: maximum concentrations of *Alternaria* toxins in quantified samples.

**Table 3 toxins-15-00201-t003:** MS/MS parameters for analyzing *Alternaria* toxins.

Compound	Structure	Retention Time (min)	Precursor Ion (*m*/*z*)	Product Ion (*m*/*z*)	Q1 Pre Bias (v)	CE (v)	Q3 Pre Bias (v)
ALS	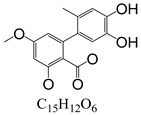	1.77	288.80	230.10 * 245.10	21.0 21.0	21 15	15.0 11.0
ALT	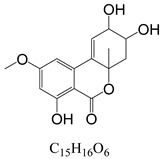	1.73	293.00	257.10 * 275.10	−15.0 −15.0	−16.0 −9.0	−30.0 −19.0
AME	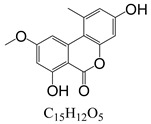	1.95	273.00	128.10 * 258.00	−15.0 −15.0	−46.0 −27.0	−25.0 −18.0
AOH	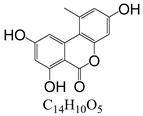	1.80	259.00	185.10 * 213.10	−14.0 −14.0	−32.0 −26.0	−12.0 −22.0
TeA	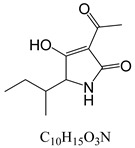	1.80	198.10	125.00 * 153.10	−11.0 −11.0	−17.0 −14.0	−24.0 −16.0
TEN	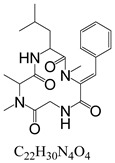	2.12	415.20	199.20 * 312.20	−13.0 −12.0	−15.0 −23.0	−12.0 −12.0

*: Quantitative ion.

## Data Availability

Not applicable.

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
