# Peer review of "Development of Acid Hydrolysis-Based UPLC–MS/MS Method for Determination of Alternaria Toxins and Its Application in the Occurrence Assessment in Solanaceous Vegetables and Their Products"

_toxins, 2023, doi:10.3390/toxins15030201_

Round 1

Reviewer 1 Report

Development of Acid Hydrolysis-Based UPLC–MS/MS Method for Determination of Alternaria Toxins and its Application in the Occurrence Assessment in Solanaceous Vegetables and their Products

The development of the extraction and purification steps are properly described, some issues should be clarified.

Lines 293-294: specify that acetonitrile is added after heating at 80°C.

Lines 298-300: was the extract diluted with water before clean up? No washing step was carried out?

Line 301. Was the final extract sample redissolved in acetonitrile or in mobile phase?

Table 3: the chemical structures are not completed.

Figure 3: change recovery (%) in y-axis with Matrix effect (%).

Regarding the surveys in solanaceous vegetables and vegetables finished products, the concentrations processing factors should be considered. For example, dry matter of tomatoes is 5-6%, while in ketchup is 28-30%; this difference could explain the higher levels found in finished products.

Finally, more and recent citations on alternariols in tomato products should be added and discussed.

Reviewer 2 Report

The main question addressed by the research is the development and validations of acid hydrolysis-based HPLC method for determination of Alternaria Toxins. The authors applied it in the quantifications of toxins in Solanaceous vegetables and their products, which list are very long and consumed worldwide.

The researchers applied modern analytical methods and techniques like solid-phase extraction (SPE) and ultrahigh-performance liquid chromatography-tandem mass spectrometry (UPLC– MS/MS). The study subject is actual. The references cited are relevant, and the most of them are from the last 5-10 years. Тhe article makes a significant contribution to food quality and safety control and ecological assessment of agricultural products in China. The authors followed the validation instructions, and the method meets the criteria for an quantification method set by the EU. The validation of the method includes an assessment of the matrix effect, which is very important considering the complexity of biological matrices.

There were collected a lot of analytical results and quantified of mycotoxins in lot of products sampled from the market.

The paper is well written, the text is clear and easy to read.

I don’t have remarks and suggestions for corrections to the authors.

Author Response

Thank you very much for such a high evaluation of the article of “Development of Acid Hydrolysis-Based UPLC–MS/MS Method for Determination of Alternaria Toxins and its Application in the Occurrence Assessment in Solanaceous Vegetables and their Products”. Thanks again.

Reviewer 3 Report

Review of the manuscript entitled "Development of Acid Hydrolysis-Based UPLC–MS/MS Method 2 for Determination of Alternaria Toxins and its Application in 3 the Occurrence Assessment in Solanaceous Vegetables and 4 their Products".

A number of analytical methods have been published on Alternaria topic so far. A section on method comparison to existing methods would be welcome. Several recently published methods are not in the reference list:

Separations 20229(3), 70; https://doi.org/10.3390/separations9030070

Papers like this are missing

My main concern is the acidic mobile phase used by the authors. Under acidic conditions TEA has distorted and unreproducible peak. That is why the authors did not present chromatogram, I believe. I would like to see chromatograms.

In the study, the authors must test the alkaline eluent composition.

The SPE purification in its presented form is not so useful. I think it is not necessary because the cartridge did not retain the compounds. What is the merit of SPE if the sample solvent is acetonitrile? Did anything improve with it?

The sample preparation is a modified QuEChERS.

Round 2

Reviewer 1 Report

Accept in current form

Author Response

Thank you very much for your review of my article.

Reviewer 3 Report

Some papers about pre-column derivatization of TEA are missing:

Asam et al. 2013, J. Chromatogr. A

Tölgyesi et al. 2015, Food  Addit. Cont. A

The chromatogram needs to be included in the paper.
